# New Insights to the Crosstalk between Vascular and Bone Tissue in Chronic Kidney Disease–Mineral and Bone Disorder

**DOI:** 10.3390/metabo11120849

**Published:** 2021-12-07

**Authors:** Maria L. Mace, Søren Egstrand, Marya Morevati, Klaus Olgaard, Ewa Lewin

**Affiliations:** 1Department of Nephrology, Rigshospitalet, University of Copenhagen, 2100 Copenhagen, Denmark; soeren.egstrand.01@regionh.dk (S.E.); marya.morevati@regionh.dk (M.M.); olgaard@rh.dk (K.O.); ewa.lewin@regionh.dk (E.L.); 2Department of Nephrology, Herlev Hospital, University of Copenhagen, 2100 Copenhagen, Denmark

**Keywords:** uremic vasculopathy, vascular calcification, renal osteodystrophy, tissue crosstalk, Wnt pathway, sclerostin, dickkopf-1, TGF-β signaling, activin A

## Abstract

Vasculature plays a key role in bone development and the maintenance of bone tissue throughout life. The two organ systems are not only linked in normal physiology, but also in pathophysiological conditions. The chronic kidney disease–mineral and bone disorder (CKD-MBD) is still the most serious complication to CKD, resulting in increased morbidity and mortality. Current treatment therapies aimed at the phosphate retention and parathyroid hormone disturbances fail to reduce the high cardiovascular mortality in CKD patients, underlining the importance of other factors in the complex syndrome. This review will focus on vascular disease and its interplay with bone disorders in CKD. It will present the very late data showing a direct effect of vascular calcification on bone metabolism, indicating a vascular-bone tissue crosstalk in CKD. The calcified vasculature not only suffers from the systemic effects of CKD but seems to be an active player in the CKD-MBD syndrome impairing bone metabolism and might be a novel target for treatment and prevention.

## 1. Introduction

What were once seen as merely structural components in the human body, the view on the vascular and skeletal systems has changed to represent organs with multiple regulatory functions. In addition to the core tasks of delivering oxygen and nutrients, and removal of waste, as well as providing posture and protection of internal organs, these two tissues secrete several factors with local and systemic effects. Research has identified vascular and bone cells as important actors in the homeostasis of the cardiovascular system, inflammation, immune system, energy balance, and mineral handling [1,2]. This is in line with other research areas, where complex interrelations between different organs and across organ systems are continuously being demonstrated [3,4,5,6,7,8].

The close relation between the vasculature and bone is illustrated by the highly vascularized bone tissue, and the important role of the microcirculation in bone development (modeling) and in the continuous cycle of remodeling, where bone is replaced by balanced bone resorption followed by bone formation [9]. A connection between the bone tissue and the peripheral circulatory system has also been demonstrated, at least in pathophysiological conditions. The observation of simultaneous vascular calcification and disturbed bone metabolism can be found in several conditions such as aging, diabetes mellitus, osteoporosis, and rarer bone diseases [10,11,12,13]. Nonetheless, it is especially pronounced in chronic kidney disease (CKD) [14]. Even though this association between the vascular and skeletal system has been known for several decades, the mechanistic link is still only sparsely understood [15,16].

Cardiovascular disease is still the most serious complication to CKD, causing increased morbidity and mortality [17,18,19]. The detrimental effects of CKD on the cardiovascular system is coupled to the concurrent disturbances in the mineral and bone balance and therefore it is part of the CKD-mineral and bone disorder (CKD-MBD) syndrome [19]. The extensive research into CKD-MBD has improved the understanding of the complex pathological mechanisms, although this has not yet resolved in improved treatment of CKD-MBD in a clinical setting [18,20]. This review will focus on vascular disease and vascular–bone tissue crosstalk in CKD. We propose a change in the current view on the role of vascular calcification in CKD-MBD.

## 2. Complex Paracrine Interplay between Bone Vasculature Cells and Bone Cells in the Development and Maintenance of Bone Tissue

During embryogenesis, bone develops simultaneously with blood vessels as a result of a complex interplay of bone-derived factors (osteokines) and vascular-derived factors (angiokines) in the microenvironment [21]. Still, the coupling of osteogenesis and angiogenesis during bone modeling is only sparsely understood at the molecular level. In the adult organism, the complex vascular network of bone illustrates the intimate spatial relation between the two tissues, and the vasculature continues to play an important role in bone remodeling and recovery after fractures. Bone tissue is maintained throughout life through the balanced bone resorption and bone formation. Fractured bone is replaced with new bone formation unlike other injured tissues, which are mainly replaced with fibrotic tissue. The endothelial cells should be seen as an integral part of bone tissue as these cells secrete factors with significant effects on both modeling and remodeling of bone. As such, paracrine regulatory loops exit between angiokines and osteokines in the bone microcirculation [22,23].

It is well known that the endothelium regulates the tonus of the vascular smooth muscle cell by nitric oxide (NO), prostaglandins, and endothelin-1 [24,25]. In bone, it has been proposed that NO also functions as a regulator of osteoblast differentiation and a mechanoresponsive mediator in the endothelial–bone cell crosstalk [26,27,28,29,30]. Endothelial-derived prostaglandin has also been proposed to promote osteoblast differentiation [22,31,32]. Endothelin-1 is associated with proliferation and differentiation of osteoprogenitor cells in bone tissue [33].

In addition, the bone vessel endothelial cells secrete several factors demonstrated to have direct effect on bone metabolism. The tumor necrosis factor (TNF) superfamily includes receptor activator of nuclear factor-κβ (RANK), RANK ligand (RANKL), and the endogenous antagonist receptor osteoprotegerin (OPG). The RANKL-RANK-OPG regulatory system is a fundamental part of bone homeostasis and represents osteoblast/osteocyte regulation of osteoclast activity [34,35]. Adding to the complexity of cell crosstalk in bone turnover, the endothelium also secretes RANKL and OPG, this is thought to regulate bone turnover in response to other stimuli such as inflammatory cytokines and vascular endothelial growth factor (VEGF) [22,36]. Bone morphogenetic protein 2 (BMP2) promotes bone formation and mineralization through a cascade of different cellular mechanisms during embryonic and post embryonic development [37]. In response to hypoxia or VEGF, the endothelial cells upregulate BMP2 expression and increase its secretion [38]. Another proposed important signal pathway is the Delta-like 4-Notch-Noggin, where noggin is secreted by the endothelial cells and promotes recruitment and differentiation of osteoprogenitor cells [22,39]. The endothelial pleiotrophin stimulates osteoblast recruitment and facilitates osteoblast attachment to the bone matrix [40]. Although several angiokines affecting bone development and homeostasis have been identified, further studies are needed to fully understand these complex cell crosstalks in normal physiology and pathophysiological conditions.

Osteolinage cells (osteoblast, osteocyte, bone lining cell, and osteoclast) secrete osteokines with paracrine effect on the vasculature. In bone modeling and remodeling, increased blood supply is observed. VEGF-A is secreted by hypertrophic chondrocytes and osteoblast-lineage cells and stimulates angiogenesis [41,42]. Crosstalk between VEGF and endothelin-1 has been demonstrated, illustrating one molecular cell communication loop between endothelial and bone cells [43]. Osteocalcin is the most abundant noncollagenous protein in the extracellular matrix of bone. It is secreted by osteoblast during bone formation but is shown to have multiple extraskeletal functions on glucose and energy metabolism, reproduction, and cognitive function [44,45]. In addition to play a role in bone development and mineral deposition (not fully understood), osteocalcin also has been shown to stimulate angiogenesis and to upregulate NO signaling in endothelial cells [46,47]. The osteoblast also secretes slit homolog 3 protein (SLIT3), which couple bone formation to angiogenesis [48]. The pre-osteoclast and mature osteoclast secrete the platelet-derived growth factor-BB (PDGF-BB), which is thought to stimulate vessel formation and recruit pre-osteoblast cells to the resorption site [49,50]. Recently, Santhanam et al., showed that osteoclast-derived PDGF-BB mediated peripheral artery disease in aged mice [51]. These results add to the understanding of the link between the vascular and skeletal system, but they also demonstrate the importance of the fine balance of secretion of osteokines. The resorption cells also secrete the matrix metalloproteases (MMPs), which are enzymes that break down the structural components of the extracellular matrix. Moreover, they stimulate angiogenesis [52,53].

The complex crosstalk between bone vascular cells and bone cells in the microcirculation of bone is essential for normal bone development, bone repair after fracture, and maintenance of bone tissue throughout life. The bone microcirculation and cell communication are most likely altered across the stages of CKD. However, this has been less studied [22,54].

## 3. Dramatic Changes in the Vasculature in CKD

CKD has deleterious systemic effects on the viability and function of several organs namely the cardio-vascular system, bone metabolism, immune system, muscle strength, energy balance, fertility, and cognitive functions. Furthermore, increasing evidence has shown that CKD is a condition of accelerated aging [55]. Of all these many complications to CKD, the CKD-MBD stands out as the most serious one affecting morbidity and mortality [17,18]. The paradigm of CKD-MBD was coined to include the complex interrelations of the pathological processes in the cardiovascular and skeletal systems as a result of reduced kidney function [19]. At the time, the disturbances in the mineral balance and bone turnover were seen as key factors in the development and progression of vascular calcification.

Several epidemiological studies report an association between plasma phosphate and increased cardiovascular disease and mortality in dialysis and pre-dialysis CKD population as well as in the general population [56,57,58,59,60,61]. The observational studies are also backed up by experimental studies illustrating a negative impact on the organism by phosphate overload [62,63,64,65,66,67]. More specifically, Li et al. showed in in vitro studies that extracellular phosphate uptake via the type III sodium phosphate co-transporter (Pit1) induced the change in the phenotype of VSMC to a bone-like cell. As such, the transformation of a contractile cell to a secretory cell, which expresses markers of osteoblastic lineage cell [68]. Expression of these markers: osteocalcin, alkaline phosphatase, BMPs, osteopontin and the osteoblast specific transcription factors runt-related transcription factor 2 (RUNX2), and osterix have also been demonstrated in the vasculature in CKD patients [69,70]. Furthermore, Zelt et al., infused radioactive labelled phosphate (^33^PO_4_) and found a great difference in buffering capacity and distribution of phosphate in adenine treated rat (CKD model). All the larger arteries had significantly higher phosphate uptake, even above the distribution in bone of CKD rats with severe vascular calcification [71]. The perturbations that arise during the development of progressive CKD are both primary and adaptive. Although plasma phosphate is well compensated until advance CKD, adaptive responses are detectable much earlier. As such, the disturbances in the mineral homeostasis start early in kidney disease with a reduction of α-Klotho and an increase in fibroblast growth factor 23 (FGF23), this is followed by a downregulation of the active vitamin D metabolite 1,25 dihydroxy vitamin D_3_ (1,25 vitamin D). Subsequently, plasma levels of parathyroid hormone (PTH) increase due to low calcium, phosphate retention, skeletal resistance to PTH signaling, and abnormalities in the regulatory feedback loops with FGF23 and 1,25 vitamin D [72].

Recent research has identified several factors in the uremic condition to be involved in vascular disease and development of vascular calcification such as uremic toxins, reactive oxygen species, DNA damage, cell senescence, inflammation, loss of local and system calcification inhibitors, secondary calciprotein particles (CPPs), and kidney-derived injury factors [73,74,75,76]. Our group has studied the uremic vasculopathy in different experimental models of kidney disease [66,77,78,79,80,81]. Recently, we have reviewed the role of the disturbed circadian clock in uremic vasculopathy and showed that it is a new important factor in the pathogenesis [81,82]. Although many pro-calcifying mechanisms have been identified, the initial nidus for calcium phosphate precipitation in the vessels is still not known.

## 4. Endothelial Dysfunction and Development of Atherosclerosis in CKD

The vascular endothelium coats the interior walls of arteries, capillaries, and veins, resulting in a large surface area. The morphology and function of the endothelial cells varies across the vascular bed, suitable for blood flow and the optimal function of the respective tissue/organ. It is a highly dynamic tissue that regulates its environment in response to external stimuli by regulating blood flow through secretion of vasodilating or vasoconstricting substances affecting the tonus of the VSMCs. In response to stress or disease, the endothelial cells can secrete reactive oxygen species and arachidonic acid, and in this way modulate vessel tonus. Various functions of the endothelial cells are pivotal in response to inflammation. These cover vasodilation, increasing the permeability of the vessel allowing cells and substance transport into the area, stimulating leukocyte migration, and changing stimuli for coagulation and thrombus formation [1].

The luminal membrane is directly exposed for uremic toxins and other disturbances caused by CKD. Changes in the glycocalyx (the protective layer of glycoproteins and proteoglycan, which reduces the endothelium’s contact with blood cells and macromolecules) has been reported in CKD patients [83]. The damage to the endothelium in CKD is most likely caused by hypertension, the state of low chronic inflammation, uremic toxin retention, dyslipidemia, and the disturbances in the mineral balance. The response to uremia involves endothelial dysfunction, impaired endothelial repair, reduced nitric oxide availability and oxidative stress. Endothelial dysfunction (also named endothel activation) describes the change in function of endothelial cells going from a steady state to a procoagulant, pro-adhesive and vasoconstricting properties [84]. All these factors trigger the development of atherosclerosis, where endothelial damage is the initial step in the pathogenesis. Followed by migration of and development of monocyte/macrophage into foam cells and deposition of lipoproteins the intima layer of the arteries. The atherosclerotic plaque is occupied with proliferating VSCMs, a necrotic core develops, which subsequently calcifies. Atherosclerosis is commonly seen in CKD patients [85,86].

## 5. Endothelial Dysfunction in CKD Vascular Calcification—An Aspect of Osteomimicry?

The phenotypic change of endothelial to mesenchymal transition (EndMT) is described in CKD, where the endothelial cell changes into a mesenchymal stem-like cell and is able to further differentiate into multiple cell lineages such as fibroblast, myofibroblast, and also bone-like cell [87]. We analyzed the dramatic genetic shift in the transcriptional profile of the severely calcified aorta from 5/6 nephrectomized rats treated with high phosphate diet and calcitriol (CKD-induced vascular calcification model) using RNA deep sequencing technique (RNA seq) (Table 1).

We found several alterations in the known angiokines and osteokines used for cell communication in the microcirculation in the bone. There was an overall increase in endothelial-derived factors known to stimulate bone formation (Table 1) [66]. Therefore, we speculate whether the arterial endothelial cell makes a phenotypic shift to a more bone-like vascular endothelial cell phenotype, which communicates with the de-differentiated VSMC/vascular bone-like cell, and hereby stimulates vascular bone formation in CKD as a kind of osteomimicry (Figure 1). This hypothesis needs more throughout examination using more specific techniques that can differentiate the origin of signal molecules in the calcified tissue and evaluate whether there is a direct pathological signaling from the endothelium to the VSMCs/bone-like vascular cells.

We also found several osteokines altered in the calcified vessel (Table 1) [66]. Again, it represents the calcification process in the medial layer of the artery. However, it could also indicate a pathological cell communication in the calcifying vessel, where the vascular bone-like cell communicates with the endothelium similarly to the known cell communication in bone (Figure 1). It may be worth looking further into whether the known signal molecules in normal cell communication in bone modeling and remodeling, also play a role in the pathophysiological calcification of vessels. In CKD, identification of signal molecules between the vascular endothelial cell and VSMC may improve the understanding of the pathogenesis of uremic vasculopathy. Hopefully, identification of early signaling or signaling leading to progression of the calcification process could represent a treatment target for vascular disease in CKD.

## 6. Development of Vascular Medial Calcification in CKD

Calcification of the medial layer of arteries is characteristic for metabolic diseases and especially CKD. Unlike focal calcification of atherosclerotic plaques, the medial calcification runs longitudinal through the vessels impairing the elastic properties as well as the dynamics of vessel’s tonus. The vascular medial calcification is mainly seen in the elastic and larger arteries, resulting in increased workload for the heart, which eventually develops left ventricular hypertrophy, reduced cardiac perfusion, and heart failure. Heart failure is commonly seen in CKD patients [18]. Induction of the transcription factors RUNX2, osterix, and BMP2 expression in the VSMCs seem to be one of the initial events in the pathological process. This is followed by expression of several genes normally expressed in bone tissue. The VSMC and bone forming cells belong to the same mesenchymal stem cell line. Unlike cardiac and skeletal muscle fibers, which are terminally differentiated cells, the VSMC keeps the ability to dedifferentiate [88]. Activation of Wnt pathway has been proposed to generate this shift in cell characteristics, and also to induce proliferation of the VSMCs [89]. Although many factors in the uremic condition are pro-calcifying, the initial stimulus for triggering the cascade of events leading to calcified vessels is not known. Other cells have also been proposed to be involved in generating the bone-like extracellular matrix, these include pericytes, adventitial myofibroblast and migrating cells [81]. Still, the current opinion is that the dedifferentiated VSMCs are central to the pathogenesis. Therefore, the shift in VSMC phenotype to a bone-like secretory cell and the change of the extracellular matrix are key events in the pathogenesis of vascular medial calcification. Maybe this ability of the VSMC to dedifferentiate is the key explanation for the observation that extra-skeletal calcification is mainly observed in the cardiovascular system.

## 7. New Concept of the Biological Systems Pathology in CKD-MBD

The classical understanding of the multiple systemic effects of CKD has been explained by a reduction of glomerular filtration, resulting in an accumulation of uremic toxins and waste products, the low grade of chronic inflammation, and the hormone disturbances related to erythropoiesis, blood pressure, mineral balance, and bone turnover. However, a new paradigm for CKD-MBD was recently proposed by Hruska et al. that kidney injury-derived factors have systemic effects on the vasculature and bone in CKD [90]. The novel theory is based on the observation of activation of development programs involved in nephrogenesis are induced by disease processes in the kidney. These development programs are involved in renal repair or fibrosis processes [90]. One principal activated system is the Wnt pathway, which regulates numerous biological functions during embryogenesis such as cell differentiation, cell polarity, proliferation, and apoptosis. It consists of several ligands, frizzle transmembrane receptors and co-receptors with activation of different intracellular signaling pathways that broadly intertwines with other signal pathways, such as TGF-β and BMP superfamily. Broadly, Wnt signaling is divided into Wnt/β-catenin canonical and non-canonical signaling, the latter includes several receptors and downstream signaling. The response to Wnt ligands also depends on the cell type. Active Wnt signaling is always coupled with expression of Wnt inhibitors, as these factors are important to balance the powerful Wnt signaling [91]. Whereas Wnt ligands are predominantly autocrine/paracrine factors, Wnt inhibitors are also circulating molecules and found increased in CKD [92,93,94]. Wnt pathway is primarily active during embryogenesis in most organs; however, active Wnt signaling is maintained in bone and is essential for normal maintenance of bone tissue [95]. Therefore, the bone tissue may be vulnerable towards activation of Wnt pathway in injured tissue if the concomitant induction of Wnt inhibitors reaches the circulation, as they theoretically could interfere with the active Wnt signaling in bone.

Another system activated in kidney disease is the TGF-β superfamily, well known for its potent regulation of extracellular matrix production and the importance of the fine balance of its activity [96]. One of its members, activin A, has been studied in relation to CKD-MBD. In CKD circulating levels of activin A are increased and the diurnal variation in plasma activin A is abrogated [97]. Activin A is a homodimer composed of two inhibin β A gene subunits (25 KDa) and only the mature protein signals through serine/threonine kinase pathway as other TGF-β superfamily members, with a highest affinity for activin receptor type 11A (ActR11a), less ActR11b. Activation of ActR11a results in phosphorylation of type 1 activin receptor-like kinase 4 (ALK4), which further phosphorylates Smad2 and Smad3, they bind Smad4 creating Smad 2,3,4 complex which enter nucleus and alter gene expression [98]. First, activin A was identified as an FSH-stimulating factor isolated from the porcine follicular fluid [99,100]. Later, it was shown to be involved in multiple biological processes during embryogenesis including development of the kidney, cardiovascular, and skeletal system. Activin A signaling remains active in the adult homeostasis in relation to the reproductive system but also active in several other organs where the function is not as well characterized. Activin A and ActR11a are widely expressed in the organism [98]. In tissue injury, including kidney injury, activin A is rapidly upregulated and has profibrotic function [79]. As such, inhibition of its signaling has been shown to improve kidney injury in animal models [101,102]. In bone, activin A is secreted by osteolineage cells and found abundantly in the extracellular matrix. Although it seems to have multiple effects in bone and affect several cell types, the overall net effect has been proposed to be a negative regulator of bone formation [103].

## 8. New Factors Identified to Play a Role in Vascular Disease in CKD

More lines of evidence are supporting the notion of an important role of these developmental factors in CKD-MBD [79,101,104]. Our group expanded the link between induction of activin A in renal injury and increased plasma levels in kidney disease by demonstrating a high secretion of activin A from the injured kidney in a model of obstructive nephropathy (unilateral ureter obstruction) [105]. We also found that kidney injury early affects the vasculature. In the model of unilateral ureter obstruction, we found that the Wnt inhibitor sclerostin was increased by 4-fold in the aorta after only 15 days, indicating early uremic vasculopathy [79]. Since normal kidney filtration was maintained by the contralateral kidney, these interesting results support the notion that local injury factors have systemic effects independent of reduced glomerular filtration. Still, more studies are needed to characterize this early signaling to the vasculature from the injured kidney. Up to now, research has focused on CKD’s detrimental effect on the cardiovascular system. By studying the effect of AKI on the vasculature and bone, it may elucidate some of the initial events in uremic vasculopathy and renal osteodystrophy.

Even though established vascular calcification persists after normalization of kidney function, some of the changes in CKD vascular disease are dynamic and may represent a target for treatment. Treatment with BMP7 has been shown to reduce plasma phosphate and aorta calcium content in an animal model of low turnover renal osteodystrophy (combined background of metabolic disturbance and CKD) [106,107]. Our group treated 5/6 nephrectomized rats with BMP7 and found some amelioration of gene changes related to fibrosis and epithelial–mesenchymal transition after 4 weeks of treatment [78]. Still, established arterial media calcification and genes related to the osteogenic transition were unchanged. The presence of calcification of the medial layer was accompanied with greater changes in gene expression related to the fibrosis process and Wnt pathway [77].

Activation and changes in the Wnt signaling pathway are central to the process of uremic vascular disease. In our molecular pathways and gene ontology cluster analysis of the genetic shift in the calcified aorta from 5/6 nephrectomized rats, we found large changes in genes related to Wnt and TGF-β pathways, remodeling of extracellular matrix and bone formation [66]. Another interesting finding of the study was identification of a strong upregulation of potentially circulating factors related to Wnt and TGF-β signaling pathways. In the top ten most upregulated genes, we found Sost coding for the canonical Wnt/β-catenin inhibitor sclerostin, which is normally expressed in bone tissue by the osteocyte. The decoy receptor for Wnt ligands secreted frizzle related protein 4 (SFRP4) was also highly upregulated [66]. Both signal molecules are found increased in the circulation in CKD. These findings suggested that the calcification process in the arteries may not only have local effect, but also produces factors which potentially can reach the circulation. If sclerostin and SFRP4 are secreted by calcified vessels, they could potentially inhibit Wnt signaling in bone and impair bone formation. The TGF-β member activin A was also highly upregulated in the calcified aorta and so this cytokine may also have an off target negative effect on bone metabolism [66]. Therefore, we have studied this novel hypothesis of a direct signal from the calcified vasculature to bone tissue.

## 9. Calcified Vasculature Affects Bone Metabolism

Since several signal molecules induced in the vasculature in the process of calcification are also found at higher levels in plasma in CKD, we speculated whether these factors were secreted by the vessel and had endocrine effect on bone, thereby linking the dysfunction of the two tissues. In order to examine our hypothesis of a vascular–bone tissue crosstalk, we developed a novel model of aorta transplantation where the calcified aorta from 5/6 nephrectomized rats (treated with a high phosphate diet and calcitriol) was transplanted into a normal recipient rat (Figure 2) [80]. In this new model, other factors in the uremic condition are excluded, and only the presence of a calcified graft remains in a healthy rat. This way, we were able to study the impact of uremic vascular calcification on bone metabolism. Due to the isogenic status of the inbred dark Aguti rats used in the study, transplantation can be performed without the use of immunosuppressive medication, which otherwise could affect bone turnover [108,109]. The bone of recipients transplanted with a calcified graft from a CKD rat showed lower trabecular tissue mineral density in the µCT analysis four weeks after transplantation. In the histological sections, lower osteoid area was found, suggesting lower bone formation. Bone volume, trabecular number, and thickness were similar between the transplanted rats. Gene analysis of bone by qPCR showed an effect on several markers of bone formation, mineralization, and resorption. These results suggested that vascular calcification has a direct impact on bone mineralization, and the follow up period of four weeks may have been too short to detect the effect on bone formation and resorption on bone morphology. The gene analysis also showed a specific upregulation of the gene Sost, coding for the canonical Wnt inhibitor sclerostin, and downregulation of Wnt target genes. These novel results indicated not only that vascular calcification inhibited the anabolic Wnt pathway in bone, but it also affected several pathways representing a complex effect on bone metabolism, suggesting several factors secreted by the uremic calcified aorta (Figure 2) [80].

We have explored the hypothesis on secretion of signal factors and a direct signaling from the vasculature to bone further in elaborated in vitro experiments. The aorta from normal rats and the calcified aorta from CKD rats were incubated ex vivo to detect possible secretions of signal molecules. We found that the calcified aorta secreted high amounts of sclerostin, illustrating that the activation of Wnt pathway in the calcifying vessel also results in secretion of sclerostin into the circulation (Figure 3) [80]. We also found secretion of the Wnt inhibitor dickkopf 1 (Dkk1) (unpublished data). Our findings support that several studies have reported an association between calcified coronary and/or larger arteries and higher plasma levels of sclerostin in CKD patients [110,111,112], as the link may be secretion of sclerostin from the calcified vessels. There is some controversy as to whether circulating Wnt inhibitors exert a direct effect on bone metabolism, partly due to the discrepancy between plasma levels of sclerostin and bone BMD. Even though sclerostin is a negative regulator of bone formation, a positive correlation between plasma levels of sclerostin and bone BMD has been reported in general population and CKD cohorts (CKD3 to CKD5 dialysis) [113,114,115,116,117,118,119,120]. These results question the role of circulating sclerostin in bone turnover and suggest that sclerostin may primarily exert paracrine signaling. In bone, sclerostin and Dkk1 are important osteocyte derived factors balancing bone formation and resorption by binding LRP5/6 of Wnt receptor complex and thereby inhibiting Wnt/β-catenin. Adding recombinant sclerostin to bone cells cultures inhibits their mineralization [121]. Whether or not vascular sclerostin has same anti-mineralization effect is not clarified, its expression is reported to co-localize with calcified areas of the vessel [110]. De Maré et al. used a different model of vascular calcification by intervening on a local calcification inhibitor. As vascular calcification progressed, the plasma levels of sclerostin increased without increased bone expression of Sost gene [122]. Collectively, the studies do suggest extraskeletal sclerostin production in CKD and likely originating in uremic vasculopathy. Dkk1 is less studied. Still, a negative association between circulating Dkk1 and BMD has been reported [116,123,124]. Further studies are needed to elucidate the role of sclerostin and dkk1 in uremic vasculopathy and vascular calcification.

In ex vivo incubation of normal and uremic calcified aortas, we have also detected high secretion of activin A by the calcified aorta rings (unpublished data). As previously mentioned, this member of the TGF-β superfamily is widely expressed in the organism and has multiple functions in normal physiology and pathology. It is thought to exacerbate the fibrosis process in uremic vasculopathy and calcification [90]. In bone, it is widely distributed in the extracellular matrix. Although the exact function of activin A in normal bone turnover is not clarified, it has been demonstrated to have a pro-osteoclastic activity and increase resorption activity. Still, other inhibitory effects on bone formation cells have also been reported [103]. Adding recombinant activin A to bone cell cultures inhibits their mineralization [125].

These early data do support the notion that factors secreted by the calcified aorta affect bone cells and the existence of a direct signal from the calcified vasculature to bone tissue resulting in disturbed bone metabolism. Clarification of this vascular–bone tissue crosstalk may improve the understanding of the CKD-MBD syndrome and the common finding of vascular calcification and lower BMD in several clinical conditions.

## 10. Can the ‘Calcification Paradox’ Be Explained by the Vascular–Bone Tissue Crosstalk?

One cannot help to speculate over the divergent processes of demineralization of bone and mineralization of vessel, the so-called ‘calcification paradox’ [126]. Although pro-osteogenic factors are induced in the vascular calcification process, several inhibitors of calcification are also upregulated [66]. Therefore, the divergent processes of pro and anti-mineralization seem to be active during mineralization of the vessels. Interestingly, we do find some similarities between the calcification process in vessel and the effect of vascular calcification on bone metabolism that may give one explanation to the ‘calcification paradox’. We found that the mineralization inhibitor osteopontin was the third most upregulated and expressed gene in the calcified aorta from 5/6 nephrectomized rats [66]. Osteopontin protein has been shown to co-localize with calcified areas in vessels [127]. This may illustrate a protective mechanism for the local environment and protection against calcium-phosphate crystal deposition. In our aorta transplantation study, we found that the normal rats transplanted with a uremic calcified aorta had significant upregulation of bone osteopontin. These rats also had upregulation of another mineralization inhibitor, ANKH [80]. In our unpublished in vitro data, bone cells exposed for calcified vasculature also upregulate osteopontin and ANKH. Therefore, we speculate in the setting of CKD, whether the vascular disease triggers a mineralization defect in bone and the impaired incorporation of calcium and phosphate in bone further exacerbate their crystallization in vessels. This results in an ongoing negative spiral of de-calcification of bone and calcification of the vasculature (Figure 4). CKD patients do exhibit mineralization defects, this has generally been assigned to disturbed vitamin D metabolism. Still, it is likely that other factors are involved in mineralization defects in CKD.

## 11. Disturbances in Wnt Pathway in Renal Osteodystrophy

Bone disorders in CKD include a wide range of disturbances, from low to high turnover diseases and mineralization defects. The CKD bone disease is classified according to the histological appearance by the TMV system (bone turnover, bone mineralization, bone volume) [19]. The pathogenesis of the different subtypes of renal osteodystrophy is poorly understood on a mechanistic level. Historically, the bone disease in CKD has been attributed to the disturbances in PTH and 1,25 vitamin D activity. However, research is identifying more and more bone-derived or systemic factors affecting bone metabolism in CKD [90,128,129,130].

In our studies on the vascular–bone tissue crosstalk, we found an upregulation of bone’s expression of sclerostin and downregulation of classical Wnt target genes in normal rats transplanted with uremic calcified aortas [80]. These results suggest that the presence of uremic vascular calcification interferes with Wnt pathway in bone. Our studies add to the growing evidence of an important role of disturbed Wnt signaling in CKD bone disease. Graciolli et al. studied bone biopsy across CKD stages 2 to 5 including dialysis patients. The majority of patients suffered from low turnover condition, even at late stage CKD. Higher bone expression of sclerostin was already found at CKD stage 2, actually preceding the increase in bone’s expression of FGF23. Patients with low bone volume also had higher plasma levels of sclerostin, which also was a good predictor of low bone turnover [131]. The presence of vascular calcification was not examined in that study; however, other groups have reported that vascular calcification is present already at stage 2 CKD [132]. Whether or not, the vascular derived sclerostin affects bone metabolism at this stage needs further studying. The mechanisms leading to this early increase in bone sclerostin are not known, but may be an early signal of disturbed Wnt pathway. It also suggests that bone disturbances initiate early in CKD and may be linked with the ongoing vascular disease. The increased bone sclerostin expression may also play a role in the increased FGF23 expression, as it was demonstrated that sclerostin upregulates FGF23 [133,134]. Besides FGF23’s endocrine functions, it is proposed that FGF23 impairs mineralization and affects Wnt pathway in bone [135,136,137].

The clinical studies are backed up by experimental studies also supporting the notion of disturbances of Wnt pathway may play an important role in renal osteodystrophy. Sabbagh et al. used a model of progressive renal disease (juvenile cystic kidneys/*jck* mouse). Already at 5 weeks, where kidney parameters of creatinine and BUN are similar to wild type, the bone of the *jck* mice showed increased sclerostin and phosphorylated β-catenin (indication of inhibition of canonical Wnt β-catenin). At 5 weeks, *jck* mice have higher plasma levels of phosphate, FGF23, and 1,25 vitamin D, but bone-derived markers of bone resorption and RANK/RANKL/OPG system are similar to wild type [127]. Taken together, changes in Wnt signaling seems to be an early event in the renal osteodystrophy of this model. Experimental studies targeting Wnt inhibitors in CKD models have reported beneficial effects. Fang et al. used an antibody to neutralize Dkk1 signaling in a model of low turnover renal bone disease and found improvements in bone volume and turnover. The effect on bone was accompanied with prevention of the osteogenic shift of the VSMC and development of vascular calcification [104]. Dkk1 is less studied in CKD patients. Moe et al. treated CKD rats (with either a high or low bone turnover condition) with anti-sclerostin antibodies and found that neutralization of sclerostin only improved bone volume in the low bone turn over condition [138]. There is an increasing resistance to PTH signaling in uremia and so higher plasma levels of PTH is needed to maintain normal bone turnover and plasma calcium. As parathyroid disease progresses from diffuse to nodular hyperplasia with autonomous PTH secretion—further stimulating a high turnover disease—lower bone volume, cortical porosity, and bone marrow fibrosis develops. This is usually seen at later stages of kidney disease. At this stage, bone expression of sclerostin is lower [131]. The importance of the different players in renal osteodystrophy may shift during the course of kidney disease. The only similarity that is maintained is the impaired incorporation of calcium and phosphate in disturbed bone metabolism. A U-shape relation between bone turnover and the presence of vascular calcification is seen in CKD patients [139]. The calcified vasculature may represent a new player affecting bone metabolism along the course of declining kidney function.

## 12. Conclusions

The vascular and skeletal systems are closely connected in normal homeostasis and pathological conditions. CKD-MBD continues to be the most serious complication to CKD regarding morbidity and mortality. The identification of the direct negative impact of vascular calcification on bone metabolism adds to the complexity of the pathological processes in CKD-MBD, and so these secreted factors from the calcified vasculature may represent a treatment target related to the vascular–bone tissue crosstalk. This novel concept of pathological vascular–bone tissue crosstalk expands the understanding of vascular and bone disease in CKD and might lead to new strategies for prevention and treatment of CKD-MBD.

## Figures and Tables

**Figure 1 metabolites-11-00849-f001:**
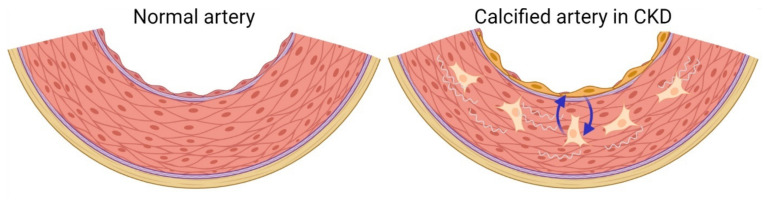
Hypothesis of pathological endothelial–vascular bone-like cell crosstalk in vascular calcification in CKD. A kind of osteomimicry? In addition to the endothelial to mesenchymal transition (EndMT) shift of the endothelial cells in CKD, the phenotypic shift of the endothelium may include expression of signals to promote vascular calcification and so mimic the endothelial cell of the bone vasculature. The de-differentiated VSMC/vascular bone-like cell may also communicate with the nearby endothelial cells, promoting vascular bone formation.

**Figure 2 metabolites-11-00849-f002:**
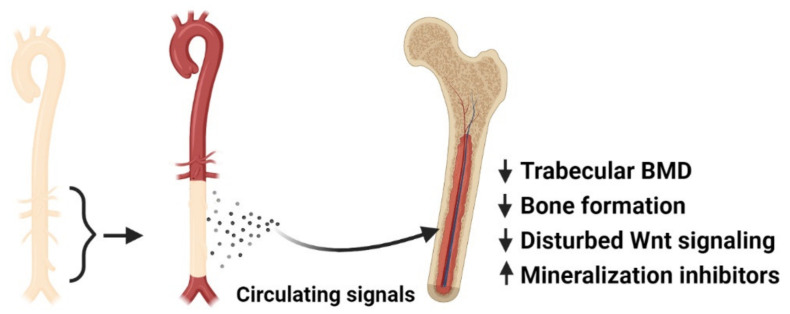
Pathological vascular calcification–bone tissue crosstalk in CKD The calcified aorta from CKD rat was transplanted into a normal isogenic recipient. The presence of the calcified aorta graft had a dramatic effect on bone of the recipient. In comparison to normal rats transplanted with a normal aorta graft, normal rats transplanted with the calcified aorta graft have lower trabecular tissue mineral density and osteoid area. These recipients of the calcified aorta graft had significant changes in expression of several markers related to bone formation, resorption, and RANKL. Sost coding for sclerostin was significantly upregulated with a downregulation of several Wnt target genes. The mineralization inhibitors osteopontin and progressive ankylosis protein homolog (ANKH) were upregulated by 3–4 fold [80].

**Figure 3 metabolites-11-00849-f003:**
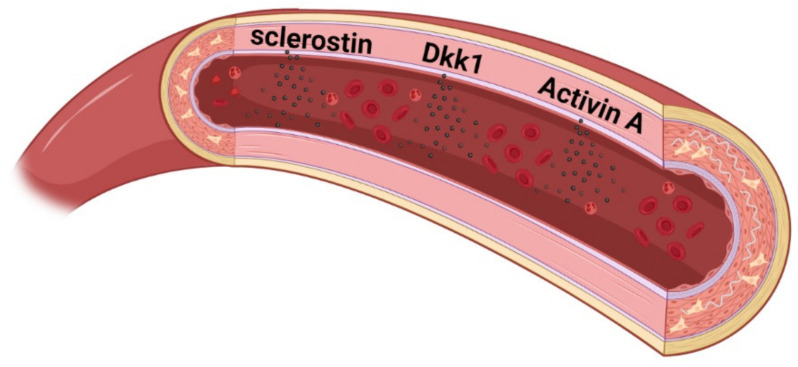
Secretion of Wnt inhibitors and TGF-β ligand from the calcified vasculature in CKD—a new pathway by which the calcified vasculature plays an active role in CKD-MBD Ex vivo incubation of the calcified aorta from CKD rats showed large secretion of sclerostin, Dkk1, and activin A [80].

**Figure 4 metabolites-11-00849-f004:**
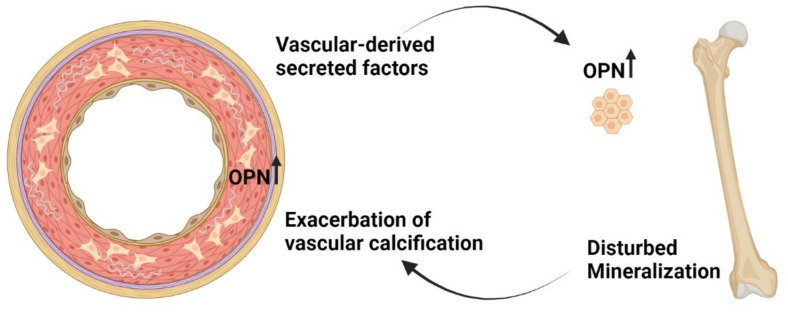
Negative spiral of de-mineralization of bone and mineralization of the vasculature. Osteopontin may play a role in the ‘calcification paradox’ during the process of vascular calcification the mineralization inhibitor osteopontin (OPN) is upregulated to be one of the most expressed genes in the diseased vessel. We propose that the stimulators of OPN expression in the vasculature also stimulate OPN expression in bone as shown in our recent study [80]. Disturbances in bone mineralization may have a negative impact on the vasculature with further precipitation of calcium-phosphate crystals. The novel concept of the pathological vascular–bone tissue crosstalk may provide a new explanation to the ‘calcification paradox’.

**Table 1 metabolites-11-00849-t001:** Aorta expression of genes coding for known molecules in the bone vascular cell and bone cell crosstalk in the normal bone homeostasis. An analysis of the genetic shift in CKD-induced vascular calcification.

Gene	Normal Aorta (Control)	CKD-Induced Calcified Aorta	Log2 Ratio CKD Aorta/Control	*p*-Value
Endothelial nitric oxide synthase (*Nos3*)	7	6	−0.17	*p* = 0.65
Prostaglandin (*Ptgs2*)	27	40	+0.53	*p* = 0.02
Endothelin 1 (*Edn1*)	1.6	3.9	+1.32	*p* = 0.01
Pecam (*Pecam1*)	89	111	+0.32	*p* = 0.23
RANK *(Tnfrsf11a)*	2.1	3.2	+0.56	*p* = 1
Osteoprotegerin *(Tnfrsf11b)*	279	179	−0.64	*p* = 0.01
BMP 2 *(Bmp2)*	1.6	3.0	+0.89	*p* = 1
Runt-related transcription factor 2 (*Runx2*)	1.3	6.0	+2.18	*p* < 0.001
Noggin (*Nog*)	0.25	1.2	+2.24	*p* = 1
Pleiotrophin (*Ptn*)	7	13	+0.78	*p* = 0.001
VEGF-A (*Vegfa*)	112	132	+0.24	*p* = 0.46
Osteopontin (*Spp1*)	443	6552	+3.88	*p* < 0.001
MMP 2 (*Mmp2*)	230	240	+0.06	*p* = 0.90
Slit homolog protein 3 (*Slit3*)	61	116	+0.93	*p* < 0.001

RNA deep sequencing (RNAseq) analysis of the aorta from normal rats and the calcified aorta from 5/6 nephrectomized rats treated with high phosphate diet and calcitriol (*n* = 5). The mRNA levels are expressed as RPKM (= reads per kilobase of exon per 1 million mapped reads) [66]. All RNA seq data are available on https://www.ncbi.nlm.nih.gov/geo/query/acc.cgi?acc=GSE146638 (accessed on the 15 October 2021).

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
