# Peer review of "New Insights to the Crosstalk between Vascular and Bone Tissue in Chronic Kidney Disease–Mineral and Bone Disorder"

_metabolites, 2021, doi:10.3390/metabo11120849_

Round 1
Reviewer 1 Report
In this paper Mace et al. present an excellent review of the crosstalk between vascular and bone tissue in CKD-MBD. The authors show the close connection between skeletal and vascular systems in health and disease, highlighting known signaling pathway alterations (like Wnt and TGFß) but also open questions in CKD-MBD and vascular calcification. Furthermore, very interesting new data are presented, adding to the manuscript's value. The work is well organized, with a multitude of references. The figures are well done. English language and stlye are fine.
Author Response
Thank you for the very nice comments.
Reviewer 2 Report
The authors have extensive experience in the study of CKD and related health disorders, including disturbances in calcium metabolism, bone mineralization, damage in the vascular system. This certainly provides the strengths of this work, since the authors not only analyze the data available in the literature on this field but also compare them with the results obtained by themselves. However, the structure of the manuscript and its filling with various information raises a number of comments that do not allow to recommend the review for publication in its current form.
A significant part of the sections is devoted to well-researched areas that have already been repeatedly analyzed in a number of reviews, including fairly recent ones. This applies to sections about endothelial dysfunction in CKD, changes in vasculature, and interaction and mutual influence of bone tissue and vascular system. Changes in mineral metabolism in CKD have also been studied in great detail. These sections do not carry a significant amount of new information and can be significantly reduced. They should only give general ideas about the problem, bringing the reader to the main subject of this review, namely new insights into the interaction of the bone and vascular systems in CKD. It is in this part of the review that its main strength and novelty lies.
Additional remarks
The section on the possibilities of treating vascular calcification needs to be expanded, or it needs to be merged with some other section. Since now, this section practically describes two approaches to treatment, and the remaining facts (e.g. calcified aorta transplantation) relate more to understanding the mechanisms of calcification and signaling, rather than to treatment possibilities.
Author Response
Thank you for your comments on our manuscript.
- Regarding the suggestion on reduction of the sections on endothelial dysfunction in CKD, changes in vasculature, and CKD-MBD.
Answer: Thank you for your comments. We have now revised the section on endothelial dysfunction in CKD and changes in vasculature in CKD, and considerably shorten the sections on mutual influence of bone tissue and vascular system referring to the recent reviews. Furthermore, the section on mineral metabolism in CKD has also been revised.
We retain the essential information so all readers, experts and others, are able to follow the new proposed paradigm in the tissue crosstalk between the vasculature and bone in CKD-MBD as well as the idea on osteomimicry in uremic vasculopathy.
- Expansion of the section on vascular calcification treatment / or merged with another section.
Answer: Thank for your suggestion. We agree that the section “Can the vascular calcification in CKD be treated?” does not fit to focus of the review. Therefore, instead of expanding the section, it is now deleted from the manuscript.
Reviewer 3 Report
In this review the authors describe in an exhaustive and updated way the interactions between the vascular system and the bone in CKD. The authors report here a recent scientific evidences showing that vascular calcification plays a leading role to in the CKD-MBD syndrome .
The tables and cartoons help to understand the complex mechanisms behind CKD-MD.
Author Response
Thank you for the nice comments.
Round 2
Reviewer 2 Report
The authors have addressed my comments and modified the text. Now the manuscript looks good.